# Predicting regional influenza epidemics with uncertainty estimation using commuting data in Japan

Taichi Murayama[ID]¹*, Nobuyuki Shimizu², Sumio Fujita[ID]², Shoko Wakamiya¹, Eiji Aramaki¹

**1** Nara Institute of Science and Technology (NAIST), Ikoma, Japan, **2** Yahoo Japan Corporation, Tokyo, Japan

* murayama.taichi.mk1@is.naist.jp

**Data Availability Statement:** All Japan influenza surveillance reports are available from the NIID (https://www.niid.go.jp/niid/ja/idwr.html) and can

## Abstract

Obtaining an accurate prediction of the number of influenza patients in specific areas is a crucial task undertaken by medical institutions. Infections (such as influenza) spread from person to person, and people are rarely confined to a single area. Therefore, creating a regional influenza prediction model should consider the flow of people between different areas. Although various regional flu prediction models have previously been proposed, they do not consider the flow of people among areas. In this study, we propose a method that can predict the geographical distribution of influenza patients using commuting data to represent the flow of people. To elucidate the complex spatial dependence relations, our model uses an extension of the graph convolutional network (GCN). Additionally, a prediction interval for medical institutions is proposed, which is suitable for cyclic time series. Subsequently, we used the weekly data of flu patients from health authorities as the ground-truth to evaluate the prediction interval and performance of influenza patient prediction in each prefecture in Japan. The results indicate that our GCN-based model, which used commuting data, considerably improved the predictive accuracy over baseline values both temporally and spatially to provide an appropriate prediction interval. The proposed model is vital in practical settings, such as in the decision making of public health authorities and addressing growth in vaccine demand and workload. This paper primarily presents a GCN as a useful means for predicting the spread of an epidemic.

## Introduction

Predicting infectious diseases is a critical task for public health authorities and industry stakeholders worldwide. Influenza (or simply flu) epidemics, representing a class of severe infectious diseases, are characterized by the widespread incidence of various symptoms, such as the sudden onset of fever, coughs, and headaches. The World Health Organization (WHO) reports that every year, 3–5 million cases of severe illness occur worldwide due to influenza, leading to 290,000–650,000 deaths annually [1]. Influenza also reduces economic productivity

be accessed following the protocol outlined in the Methods section.

**Funding:** This study was supported in part by Yahoo Japan Corporation. The funder provided support in the form of salaries for Dr. Nobuyuki Shimizu and Mr. Sumio Fujita, but did not have any additional role in the study design, data collection and analysis, decision to publish, or preparation of the manuscript. The specific roles of these authors are stated in the 'author contributions' section.

**Competing interests:** The authors have declared that funding from a commercial source, Yahoo Japan Corporation, does not alter our adherence to PLOS ONE policies on sharing data and materials.

because of employee absenteeism and sudden increase in hospital workload [2]. Such instances have motivated public health authorities to predict the consequences of influenza in different countries.

Existing influenza prediction systems must be improved to make better decisions regarding public health. First, the influenza volume should be predicted over small regions, rather than over entire countries. Second, the reliability of prediction results should be investigated. Regional influenza predictions must consider the characteristics of infectious diseases, which are mainly spread through direct contact with infected persons (contact infection) or the sneezing and coughing of infected persons, which can lead to the spread of infectious droplets in the air (droplet infection) [3, 4]. Thus, influenza tends to spread from one area to the surrounding areas through direct contact with infected persons. According to previous research, such a regional infection spreading pattern can be better modeled by considering the flow of people between regions, rather than considering spatially-adjacent relations [5–7]. Additionally, public health organizations must comprehend the degree of prediction confidence. This will stimulate a flexible response to various problems triggered by influenza epidemics.

This study aimed to develop a regional flu prediction model that incorporates the geographical flow of people and uncertainty estimation for a cyclic time series. To achieve this, we used commuting data to model the flow of people into a region from other regions. In particular, inter-regional commuting information, as shown in Fig 1(b), was used instead of regional adjacency data (AD), as shown in Fig 1(a). We incorporated influenza data and commuting data into a traffic simulation model to assess the spread of infection caused by the flow of people based on geographical relations. This study extended the use of graph convolutional neural networks (GCNs) to capture latent geographical relations using graph representation, where each node of the graph is a target region for influenza prediction, and each edge represents the commuting flow of people. GCNs capture spatial dependencies and can be easily combined with other neural models to improve prediction. It is important to show that a GCN can effectively predict the geographical distribution of influenza. We aimed to construct an infectious disease prediction system for each region.

Furthermore, we estimated the suitable uncertainty of our model's prediction using a prediction interval. This is important for decision making in terms of public health regarding factors such as vaccine demand and medical personnel allocation. Our spatiotemporal model is based on neural networks that are adopted by some epidemic prediction studies [6, 8]. However, it is difficult to estimate the prediction interval for the downside or upside of prediction points because neural networks conduct point estimation. Therefore, owing to the unknown reliability of prediction results, it becomes difficult for public health authorities to take certain decisions. To resolve such difficulties, Zhu et al. [9] presented an encoder–decoder method with an inference of prediction intervals by calculating three sources of prediction uncertainties, i.e., model uncertainty, inherent noise, and model misspecification, using Monte Carlo (MC) dropout, which was derived from the property of dropout-approximate Bayesian inference. This method appends an inference module to a trained model, without re-training it, to estimate the prediction uncertainties of the model. However, Zhu et al.'s method tends to favor a prediction interval that is much larger than adequate for non-epidemic periods (mainly in summer). This is due to the lack of consideration of the one-year periodicity in the time series of flu data, which exhibit strong seasonality (i.e., epidemic in winter and non-epidemic in summer). In brief, Zhu et al.'s method is not suitable for a time series with periodicity, such as flu data. Therefore, we extended their method to estimate a suitable prediction interval for one-year cyclic trends in time series and evaluated the effectiveness of the extended method.

The main contributions of this study can be summarized as follows:

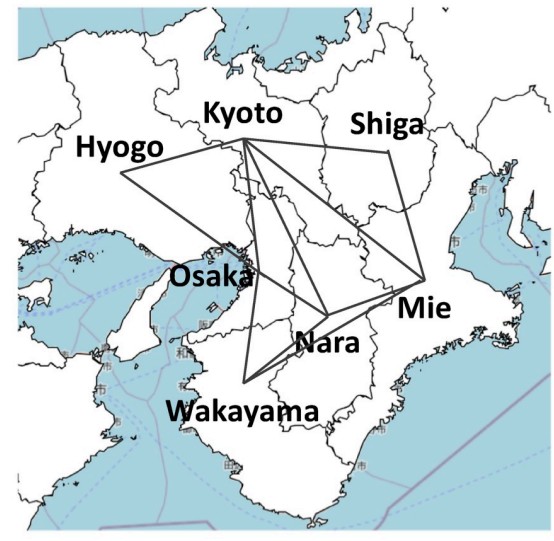

| To<br>From | Hyogo | Osaka | Kyoto | Shiga | Nara | Waka<br>yama | Mie |
|---|---|---|---|---|---|---|---|
| Hyogo | 0 | 1 | 1 | 0 | 0 | 0 | 0 |
| Osaka | 1 | 0 | 1 | 0 | 1 | 1 | 0 |
| Kyoto | 1 | 1 | 0 | 1 | 1 | 0 | 1 |
| Shiga | 0 | 0 | 1 | 0 | 0 | 0 | 1 |
| Nara | 0 | 1 | 1 | 0 | 0 | 1 | 1 |
| Waka<br>yama | 0 | 1 | 0 | 0 | 1 | 0 | 1 |
| Mie | 0 | 0 | 1 | 1 | 1 | 1 | 0 |

| To<br>From | Hyogo | Osaka | Kyoto | Shiga | Nara | Waka<br>yama | Mie |
|---|---|---|---|---|---|---|---|
| Hyogo | 0 | .3 | 0 | 0 | 0 | 0 | 0 |
| Osaka | 0 | 0 | 0 | 0 | .1 | 0 | 0 |
| Kyoto | 0 | .4 | 0 | 0 | 0 | 0 | 0 |
| Shiga | 0 | 0 | .1 | 0 | 0 | 0 | .1 |
| Nara | 0 | .5 | 0 | 0 | 0 | 0 | 0 |
| Waka<br>yama | 0 | .1 | 0 | 0 | 0 | 0 | 0 |
| Mie | 0 | 0 | 0 | 0 | 0 | 0 | 0 |

(a)Adjacency Matrix
(Adjacency Data)

(b) Weighted directed matrix
(Commuting Data)

**Fig 1.** (a) Adjacency matrix, which is undirected with no weights, has been used so far. (b) Weighted directed matrix, originating from commuting data, includes weights and directions to assess infectious disease characteristics.

1. We demonstrate that modeling the flow of people as spatial information is useful for regional flu prediction. Our spatiotemporal model aims to provide better predictions than baseline models.

2. We introduce an uncertainty estimation method for cyclic time series with real-life applications (such as the prediction of infectious diseases).

The proposed model with uncertainty estimation has important applications, including decision support for regional public health authorities in terms of vaccines and workload.

## Related work

### Influenza prediction

Influenza prediction methods can be broadly classified into three categories: compartmental model-based, statistical and time series, and machine learning. Compartmental models

include the "susceptible–infected–removed" (SIR) [10] and incidence decay with exponential adjustment [11] models. They differ from statistical and machine-learning methods as they set suitable parameters for each compartment and focus on understanding disease dynamics. Statistical and time series methods include the autoregression-integrated moving average [12] and generalized autoregressive moving average [13] methods. In particular, the autoregression with Google search (ARGO) method [14], which is based on linear regression using the input data of the Google search time series and historical influenza-like illness data, has exhibited superior results for flu forecasting [15–17]. Our GCN-based model is based on machine learning. Other examples of machine-learning methods include linear regression [18, 19], random forest [20], Gaussian process [21], and long short-term memory (LSTM) [7, 8, 22].

Resource selection for the prediction method is also an important factor in influenza prediction. Many studies have relied on user-generated content (UGC) from internet services, such as search services [12, 14, 23, 24] and social networking services [25–27]. Infectious disease surveillance conducted with online content, such as that described above, is generally described as infoveillance [28]. Currently, Google Flu Trends [24] is one of the most representative systems, which is designed to estimate the current influenza-like illness rate using related Google search terms. Signorini et al. [29] examined Twitter streams for the volume of tweets including keywords related to influenza and demonstrated the usefulness of Twitter data for tracking flu epidemics. In addition to user-generated content, many studies have used diverse resources to improve their models, such as Wikipedia [30], historical flu data [14, 31–33], and weather data [34]. Our model used historical flu data as a resource.

Moreover, our research on influenza prediction for each prefecture is related to the following studies. Senanayake et al. [5] used a kernel function based on the distance between two areas to capture spatial dependence. Wu et al. [6] used a convolutional neural network (CNN) architecture to convolve the information of surrounding areas. Liu et al. [35] used a geographically weighted regression model, which extended the ordinary linear regression model and embedded geographical location data into the regression parameters, with geographical information about hospitals, such as the number of hospitals per 10,000 population, to predict the COVID-19 situation in China. In contrast to the abovementioned studies, our study used regional commuting data to model the flow of people into a specific area. Brockmann et al. [36] attempted to capture the onset of an epidemic using data on international traffic. Wang et al. [37] extended the classic SIR model to consider the visitor transmission between any two areas to predict intra-city epidemic propagation using the traffic volumes in cities. To the best of our knowledge, our study is the first attempt to predict influenza volume in detail for a large area, i.e., the entire territory of Japan, by considering the inter-regional flow of people using machine learning.

## Spatiotemporal model

Spatiotemporal models have a long history [38] as below. Dynamical state-space models, where the current state is conditioned in the past, have also been explored [39]. The use of tensor methods to analyze epidemic data [40] and models that detect the movement of a person in a video using conditional random fields [41] are examples of spatiotemporal models. Recently, GCNs [42], which convolve the graph architecture, were used for text classification [43], image analysis [44], and molecular structure analysis [45]. Additionally, GCN models can present regional relations as graphs and capture time dependence. Previously, GCNs were studied for traffic prediction problems, such as bicycle flow [46] and traffic volume [47].

## Bayesian neural networks

Bayesian neural networks (BNNs) are derived from Bayesian methods and can incorporate uncertainty in deep learning models. The method described by Zhu et al. [9], which is the base model for our prediction interval estimation, is related to BNNs. A BNN aims to determine the posterior distribution of network parameters rather than conduct point estimation. However, it is difficult to calculate the posterior inference of deep learning models because of their complex nonlinearity and non-conjugacy characteristics. Several approximate inference methods have been proposed to address this difficulty, such as probabilistic backpropagation [48] and stochastic search [49]. Zhu's method is based on the MC dropout proposed in [50]. An important feature of MC dropout is that it can be easily applied to neural networks because it performs stochastic dropouts after passing through each learned hidden layer; further, it generates a posterior predictive distribution.

# Materials and methods

This section describes the proposed model. We propose a spatiotemporal model inspired by [9, 51] that incorporates the geographical flow of people. Moreover, the model incorporates an estimation method for influenza prediction that is suitable for a year-long cyclic time series. Our model consists of two parts: influenza prediction and uncertainty (prediction interval) estimation. Fig 2 illustrates an overview of the proposed model.

## Influenza prediction

The influenza prediction part of the model is composed of two combined modules: a GCN and a sequence-to-sequence architecture. The GCN extracts the features of various spatial relationships between observation points and captures spatial dependencies. The GCN can be easily combined with other neural networks, such as a recurrent neural network (RNN), which is useful in predicting infectious diseases [22]. The GCN can be used for feature extraction related to graph nodes. Overall, the GCN can achieve high accuracy in predicting infectious diseases. Based on the above reasons, we selected a GCN to capture spatial dependencies. Our model also employs a sequence-to-sequence architecture, which is useful for producing forecasts more than two weeks in advance. Table 1 defines the main notations used to represent the influenza prediction part of our model.

**Task definition.** The objective of influenza prediction is to predict the number of future influenza patients based on previously observed data and commuting data corresponding to $N$ regions in the network. One can use $\boldsymbol{X}^{(t)} \in \mathbb{R}^{N \times M}$ to represent $M$ epidemiology information observed from $N$ different signals at time $t$; for example, the number of influenza patients in $t$ weeks in $N$ regions of Japan. Additionally, we represent the regional network as a weighted directed graph $\mathcal{G} = (\mathcal{V}, \mathcal{E}, \boldsymbol{W})$, where $\mathcal{V}$ is a set of nodes $|\mathcal{V}| = N$, $\mathcal{E}$ is a set of edges, and $\boldsymbol{W} \in \mathbb{R}^{N \times N}$ is a weighted matrix representation, such as the constant commuting volume between regions. The influenza prediction problem aims to learn the function $f(\cdot)$ that maps $T'$ historical signals and a constant weighted matrix representation of $\mathcal{G}$ to $T$ future signals:

$$[\boldsymbol{X}^{(t-T'+1)}, \ldots, \boldsymbol{X}^{(t)}; \mathcal{G}] \xrightarrow{f(\cdot)} [\boldsymbol{X}^{(t+1)}, \ldots, \boldsymbol{X}^{(t+T)}]$$

**Diffusion graph convolutional network.** We used a diffusion GCN (DGCN), which was originally developed for traffic flow prediction by [51], where we modeled the spatial dependence of the virus spreading by applying a diffusion process, i.e., random walk on a commuting graph. Thus, the temporal dynamics of the infection spread through regions were captured

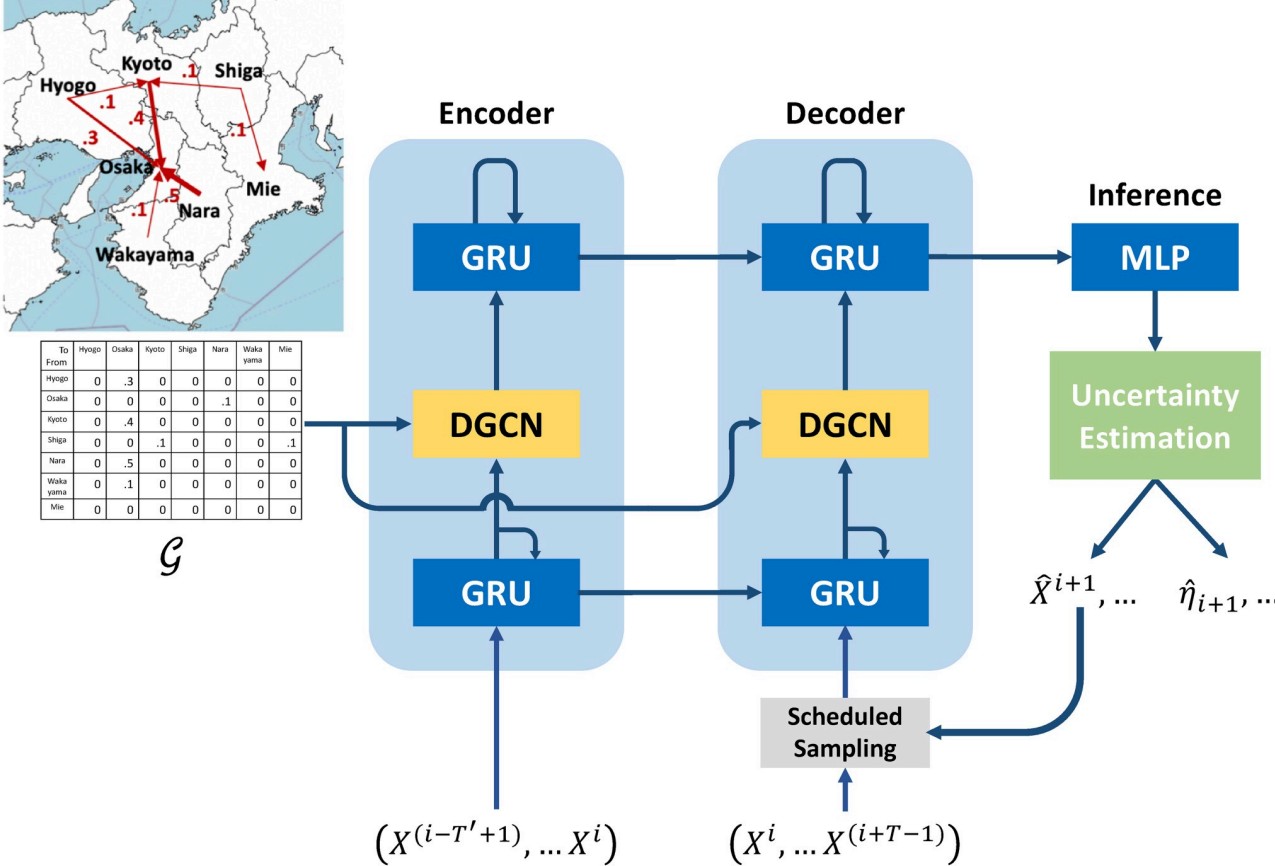

**Fig 2. Overview of our model.** The model includes sequence-to-sequence combinations of a diffusion GCN and gated recurrent unit (GRU) with uncertainty estimation. We feed the historical time series of patient numbers into the encoder. Next, we use its final states to initialize the decoder. The decoder generates a prediction from previous ground-truths or the values predicted by the model using scheduled sampling. Additionally, our model applies the predicted values to our uncertainty estimation method and then outputs the prediction interval.

by a stochastic process on the input graph $\mathcal{G}$. Intuitively, this stochastic process represents the step-by-step "flows of viruses" through regions; one day, a commuter transmits a virus to a region, and the following day, other commuters transmit the virus from this region to other regions with some probability, and so on. The transition matrix of the diffusion process is $\mathbf{D}_O^{-1}\mathbf{W}$, where $\mathbf{D}_O = diag(\mathbf{W}\,\mathbf{1})$ is the diagonal matrix of the total out-commuters from each region, and $\mathbf{1}$ denotes the all-ones vector. The stationary distribution of the diffusion process is as follows:

$$\mathbf{P} = \sum_{k=0}^{\infty}\alpha(1-\alpha)^k(\mathbf{D}_O^{-1}\mathbf{W})^k \tag{1}$$

where $k$ represents the number of diffusion steps and $\alpha \in [0, 1]$ represents the restart probability, with which the diffusion process restarts from its initial states [52, 53]. The DGCN adopts a graph diffusion convolution using the above-mentioned diffusion process in Eq (1)over an input epidemiology signal $\mathbf{X}$ and a filter $f_{\theta}$, leveraging the flows both leaving and entering each region. The signal information $X$, such as the current number of patients, is transferred from one node to its neighboring nodes with the probabilities given in the transition matrix, and the spread signal distribution can reach the above-mentioned stationary distribution after several

**Table 1. Main notations.**

| Notation | Definition or Description |
|---|---|
| **X(t)** | epidemiology information at time $t$ |
| **W** | weighted matrix |
| $\mathbf{D}_O$ | out-degree diagonal matrix |
| $\mathbf{D}_I$ | in-degree diagonal matrix |
| **Θ** | filter parameter tensor |
| **O** | output of DGCN |
| $\mathbf{H^1, H^2}$ | output of GRU |
| $\hat{X}^t$ | predicted influenza volume at time $t$ |
| $\hat{\eta}_t$ | total prediction uncertainty at time $t$ |
| $I$ | number of input features |
| $N$ | number of nodes (regions) |
| $M$ | number of input features for DGCN |
| $Q$ | number of output features for DGCN |
| $T'$ | input length |
| $T$ | output length |
| $K$ | number of diffusion steps |

steps. However, the DGCN uses only a finite $K$-step truncation of the whole diffusion process for computational efficiency. Thus, it captures the $K$-localized graph structures of $G$ as follows:

$$
\begin{aligned}
\boldsymbol{X}_{:,m} \star \mathcal{G} \boldsymbol{f}_{\boldsymbol{\theta}} &= \sum_{k=0}^{K-1} (\theta_{k,1}(\boldsymbol{D}_O^{-1}\boldsymbol{W})^k \\
&+ \theta_{k,2}(\boldsymbol{D}_I^{-1}\boldsymbol{W}^T)^k)\boldsymbol{X}_{:,m} \quad m \in \{1 \ldots M\}
\end{aligned}
\tag{2}
$$

where $\boldsymbol{\theta} \in \mathbb{R}^{K \times 2}$ are the filter parameters, and $\star \mathcal{G}$ denotes a graph convolution operation. Furthermore, $\boldsymbol{D}_O^{-1}\boldsymbol{W}$ and $\boldsymbol{D}_I^{-1}\boldsymbol{W}^T$ represent the transition matrices of the diffusion and reversed processes, respectively, when considering both flows of people. Our machine learning method uses both directions; it learns different parameters for each transition matrix. These two directions might affect the epidemic situation in regions with different strengths of impact; thus, the input graph must be directed.

However, computation of the convolution operation defined in Eq (2) may be expensive. To localize the filter and reduce the number of parameters, the first part of Eq (2), including $\boldsymbol{D}_O^{-1}\boldsymbol{W}$, can be rewritten as

$$
\sum_{k=0}^{K-1} \theta_k T_k(\boldsymbol{X}_{:,m})
\tag{3}
$$

As $T_{k+1}(x) = \boldsymbol{D}_O^{-1}\boldsymbol{W}T_k(x)$ and $\boldsymbol{D}_O^{-1}\boldsymbol{W}$ are sparse, the computational cost can be reduced by recursively computing $K$-localized convolutions [54].

Regarding the convolution operation defined in Eq (2), a diffusion convolutional layer maps $M$-dimensional features to $Q$-dimensional outputs, where $Q$ is the number of output features. The diffusion convolutional layer is described as

$$
\mathbf{O}_{:,\mathbf{q}} = \mathbf{a}\left( \sum_{\mathbf{m}=1}^{\mathbf{M}} \mathbf{X}_{:,m} \star \mathcal{G} \mathbf{f}_{\Theta_{q,m,:,:}} \right) \quad \mathbf{q} \in \{\mathbf{1} \ldots \mathbf{Q}\}
\tag{4}
$$

where $\mathbf{O} \in \mathbb{R}^{\mathbf{N} \times \mathbf{Q}}$ represents the output, $\Theta \in \mathbb{R}^{Q \times M \times K \times 2}$ consists of all $\theta$ parameters in the parameter tensor, and $\mathbf{a}$ represents the activation function (e.g., ReLU and sigmoid).

**Sequence-to-sequence architecture of GRU and DGCN.**   Our model employs a sequence-to-sequence architecture to provide forecasts more than two weeks ahead of time; these are composed of RNNs to model the temporal dependence and a GCN to model the spatial dependence. In particular, a GRU [55], which is a simple and powerful variant of an RNN, was first used. The GRU considers $\mathbf{X}^t$ and $\mathbf{H}_{t-1}$ as inputs and outputs $\mathbf{H}_t$ in accordance with the following formulae:

$$\mathbf{r}_t = \sigma(U_r \mathbf{X}_t + W_r \mathbf{H}_{t-1}) \quad \mathbf{f}_t = tanh(U_h \mathbf{X}_t + \mathbf{H}_{t-1} \odot W_h \mathbf{r}_t)$$
$$\mathbf{z}_t = \sigma(U_z \mathbf{X}_t + W_z \mathbf{H}_{t-1}) \quad \mathbf{H}_t = (1 - \mathbf{z}_t) \odot \mathbf{H}_{t-1} + \mathbf{z}_t \odot \mathbf{f}_t \tag{5}$$

where $\mathbf{z}_t$ and $\mathbf{r}_t$ represent the reset gate and update gate at time $t$, respectively. $U_z, U_r, U_h \in \mathbb{R}^{I \times M}$ and $W_z, W_r, W_h \in \mathbb{R}^{M \times M}$ are parameters for the respective gates, and $M$ is the output dimension of the GRU. We can consolidate Eq (5) as follows:

$$\mathbf{H}_t^1 = GRU(\mathbf{X}^t), \quad t \in \{(i - T' + 1), \dots, i\} \tag{6}$$

where $\mathbf{H}_t^1 \in \mathbb{R}^{N \times M}$, which is the hidden state of the GRU, is applied by the DGCN, as described in Section 4.1. We can then represent Eqs 1–4 as follows:

$$\mathbf{O}_t = DGCN(\mathbf{H}_t^1, \mathbf{W}) \tag{7}$$

The DGCN is used between the two GRU layers to achieve feature squeezing, as described in [56]. Subsequently, we apply the output of the DGCN to the second GRU layer, as follows.

$$\mathbf{H}_t^2 = GRU(\mathbf{O}_t) \tag{8}$$

where $\mathbf{H}_t^2 \in \mathbb{R}^{N \times S}$. For the inference of the influenza volume in each region, we apply the output of the second GRU layer in the decoder to the multilayer perceptron (MLP), which has two layers. Finally, $\hat{\mathbf{X}}_n^{(t+1)}$, which is the final output, represents the number of influenza patients in $n$ regions at time $t + 1$:

$$\hat{\mathbf{X}}_n^{(t+1)} = MLP(\mathbf{H}_{t,n}^2) \tag{9}$$

During training, we feed the historical time series of patient numbers into the encoder. Next, we use its final states to initialize the decoder, which generates the prediction from previous ground-truth values. However, the discrepancy between the input distribution of training and testing data can decrease the performance, as because ground-truth values are replaced by predictions generated by the model. To solve this problem, we use scheduled sampling [57], which is a process that feeds the model either ground-truth values with probability $\epsilon$ or model predictions with probability $1-\epsilon$.

## Uncertainty estimation

Our model incorporates a method to estimate the uncertainty of the model prediction, i.e., a prediction interval suitable for a cyclic time series. Estimating prediction intervals is important for public health organizations when making decisions.

However, it is difficult to apply neural networks that conduct point estimation, such as our prediction model. Therefore, we propose a method for estimating prediction intervals that are suitable for cyclic time series after explaining Zhu's method [9], which is the basis of our method.

**Algorithm 1** Inference (from [9])
**Input:** data $x^*$, encoder $g(\cdot)$, prediction network $h(\cdot)$, dropout proba-
bility $p$, number of iterations $B$
**Output:** prediction $\hat{y}^*_{mc}$, uncertainty $\eta$
  *//Model uncertainty and model misspecification*
1: $\hat{y}^*$, $\eta_1 \leftarrow$ MCdropout $(x^*, g, h, p, B)$
  *// Inherent noise*
2: **for** $x'_v$ **in** validation set $\{x'_1, \ldots, x'_V\}$ **do**
3:     $\hat{y}'_v \leftarrow h(g(x'_v))$
4: **end for**
5: $\eta_2^2 \leftarrow \frac{1}{V}\sum_{v=1}^{V} (\hat{y}'_v - y'_v)^2$
  *// Total prediction uncertainty*
6: $\eta \leftarrow \sqrt{\eta_1^2 + \eta_2^2}$
7: **return** $\hat{y}^*$, $\eta$

**Algorithm 2** Inference considering periodicity
**Input:** data $x^*_m$, time cyclic point $m$, encoder $g(\cdot)$, prediction network $h$
$(\cdot)$, dropout probability $p$, number of iterations $B$, window width $W$
**Output:** prediction $\hat{y}^*_{mc}$, uncertainty $\eta$
  *//Model uncertainty and model misspecification*
1: $\hat{y}^*$, $\eta_1 \leftarrow$ MCdropout $(x^*_m, g, h, p, B)$
  *// Inherent noise*
2: **for** $x'_w$ **in** $\{x'_{m-(W-1)/2}, \ldots, x'_{m+(W-1)/2}\}$ **do**
3:     $\hat{y}'_w \leftarrow h(g(x'_w))$
4: **end for**
5: $\eta_2^2 \leftarrow \frac{1}{W}\sum_{w=1}^{W} (\hat{y}'_w - y'_w)^2$
  *// Total prediction uncertainty*
6: $\eta \leftarrow \sqrt{\eta_1^2 + \eta_2^2}$
7: **return** $\hat{y}^*$, $\eta$

**Base method.** We describe the method proposed by Zhu et al. (referred to as Zhu's method), which provides time-series prediction and uncertainty estimation. This method quantifies the standard error $\eta$ of the prediction. Therefore, an approximate $\alpha$-level prediction interval can be constructed using $[y^* - z_{\alpha/2}\,\eta, y^* + z_{\alpha/2}\,\eta]$. Here, the model prediction $y^* = f^{\hat{W}}(x^*)$, $x^*$ is a new input, $f^{\hat{W}}(.)$ is a trained neural network, and $z_{\alpha/2}$ is the upper $\alpha/2$-quantile of the standard normal distribution. The method accounts for three sources of prediction uncertainties for quantifying the prediction standard error $\eta$: model uncertainty, inherent noise, and model misspecification.

Model uncertainty and misspecification are calculated using MC dropout, which was derived from the property of dropout-approximate Bayesian inference [50]. Specifically, MC dropout proceeds to randomly drop out each hidden unit in a model with a certain probability $p$. This stochastic feed-forward process is repeated $B$ times to obtain an output $\{\hat{y}^*_{(1)} \ldots, \hat{y}^*_{(B)}\}$. Using this output, we can approximate the model uncertainty as

$$\widehat{\mathrm{Var}}(f^W(x^*)) = \frac{1}{B}\sum_{b=1}^{B}(\hat{y}^*_{(b)} - \bar{\hat{y}}^*)^2 \tag{10}$$

where $\bar{\hat{y}}^* = \frac{1}{B}\sum_{b=1}^{B}\hat{y}^*_{(b)}$. To incorporate this uncertainty into the encoder–decoder model, we apply MC dropout to all layers in both the encoder $g$ and final prediction network $h$. Estimation of the model uncertainty and misspecification using MC dropout is described in [9].

The inherent noise $\hat{\sigma}^2$ is estimated via the residual sum of squares evaluated on an independent validation set $X' = \{x'_1, \ldots, x'_V\}$, $Y' = \{y'_1, \ldots, y'_V\}$. We estimate the inherent noise via the residual sum of squares for the validation set as we do not know the correct noise level a

priori.

$$\hat{\sigma}^2 = \frac{1}{V}\sum_{v=1}^{V}(y_v' - f\hat{W}(x_v'))^2$$

Uncertainty estimation is presented in Algorithm 1.

**Proposed method for cyclic time series.** We incorporate Zhu et al.'s uncertainty estimation method into our model for flu prediction. Fig 6(a) shows the time series of our model using Zhu's method for the Okayama prefecture. This figure shows the method applied to our model for the prediction interval. Specifically, it shows a tendency to provide a larger than necessary prediction interval in a non-epidemic period, where there is only a slight variation in the number of flu patients. This tendency, which originates from the method of calculating the inherent noise, can complicate decision making for health authorities.

The inherent noise in Algorithm 1 is assumed to be constant in all periods. However, inherent noise strongly depends on the season in a year-long periodic time series (such as the number of influenza patients). Therefore, we replace Algorithm 1 with Algorithm 2, and subsequently incorporate it into our model with this uncertainty estimation for cyclic time series. In Algorithm 2, the one-period validation set is used to calculate the inherent noise, as periodicity must be considered. In particular, we prepare the one-period validation set $X' = \{x_1', \ldots, x_M'\}$, $Y' = \{y_1', \ldots, y_M'\}$ in the time series (e.g., one-year validation set for flu prediction). Next, we calculate the inherent noise using window width $W$ of the validation set $\{x_{m-(W-1)/2}', \ldots, x_{m+(W-1)/2}'\}$ around January, when a new input is in the January data $x_m^*$. Here, $m$ is the time cyclic point, and $M$ is the number of one-period data points.

## Experiments

We evaluated the predictive capabilities of our spatiotemporal model and prediction interval estimation on the 47 prefectures of Japan. The proposed model is referred to as "GCN+Seq2seq w/ PF" hereinafter, where PF indicates that the model considers the flow of people.

We aimed to answer the following research questions:

- (RQ1) Does commuting data improve the accuracy of influenza prediction?

- (RQ2) When and in which area does our model produce good results?

- (RQ3) How effective is our uncertainty estimation method in real-world epidemic prediction?

## Datasets

**Influenza data.** We used data based on the weekly number of patients with influenza symptoms for each prefecture in Japan, as reported by the National Institute of Infectious Diseases (NIID). NIID reports aggregated information related to influenza in its weekly reports [58] to provide warnings regarding infection outbreaks. These reports are delayed by approximately seven days from the date of the original clinical reports by physicians (because of the time necessary to aggregate clinical information from different health authorities in each prefecture). We mixed all subtypes of influenza data and accumulated the number of influenza patients from the 37th week of 2012 to the 30th week of 2020. We accessed the data, which was provided fully anonymized, on 21 Oct 2020.

**Commuting data.**   Spatiotemporal models typically use adjacency and distance between observation points to model geographical information. However, human mobility is strongly linked with the transmission of infectious diseases (such as droplets and contact infections). Therefore, this study used commuting data instead of AD as geographical information (Fig 1). Adding commuting data to our model can better capture the epidemic situation in a region, which is important for public health organizations.

To consider inter-regional flows of people, we used commuting data from the 47 prefectures of Japan. The data were provided by the national census report [59] of 2015. The provided data include the single daily average numbers of commuters from one prefecture to another over all days of the weeks. We accessed the data, which was provided fully anonymized, on 21 Oct 2020. In the experiment in each year, we represented the data of each year as a graph $\mathcal{G} = (\mathcal{V}, \mathcal{E}, \mathbf{W})$ in the proposed model because the national census report only provides only the number of commuters, regardless of the year. We divided the number of commuters by the maximum number of commuters between every prefecture pair, which is known as min-max normalization to graph-edge information, such as $\mathbf{W}$. The maximum number of commuters (270,000) travels from Kanagawa prefecture to Tokyo prefecture. Moreover, 135,000 commuters travel from Osaka to Nara. The edge weight, representing commuters traveling from Osaka to Nara, is shown as 0.5 (=135,000/270,000) in the graph, after applying min-max normalization.

## Models for comparison

**Vector autoregression.**   Vector autoregression (VAR) [60] is an extension of autoregressive models that allow for more than one evolving variable. We selected observation values in all regions, which are up to $T'$ weeks before, as multiple variables. We set $T'$ as five weeks in the experiment. To make the model more robust, we adopted an L2-regularization term for training.

**LSTM.**   The LSTM model captures temporal dependence in data and preserves backpropagated error through time and layers. LSTM has been successfully used in natural language and sound signal processing [61] as well as influenza prediction [8, 22]. Specifically, LSTM has input, output, and forget gates, which are used to compute the new states in the memory cell given old values. Our baseline architecture is the same as that reported in [8].

**CNNRNN-Res.**   The CNNRNN-Res model was developed by [6] for influenza prediction. The model structure comprises three parts: a CNN to capture regional relations; an RNN to capture time dependencies; and residual links for fast training with no overfitting. The CNN uses the adjacent information of the respective regions. The residual links bypass some intermediate layers, which can mitigate overfitting [62].

**GCN+Seq2seq w/ AD.**   To validate the effectiveness of PF as spatial information compared with other geographical relations between prefectures, we used our model with AD instead of commuting data. Note that AD comprise a matrix that represents whether two regions are adjacent (1) or not (0) without a specified direction, as shown in Fig 1(a). We term this model with AD between the 47 prefectures "GCN+Seq2seq w/ AD," for contrast with the proposed"GCN+Seq2seq w/ PF" model.

**GCN+Seq2seq w/ DD.**   To validate the effectiveness of PF as spatial information, compared with other geographical relations between prefectures, we considered the distance between prefecture regions. We assume that the inter-region distance is an important factor in estimating the strength of the interaction between regions along with geographical adjacency. We prepared the data by measuring the straight-line distance between the locations of the government offices of each prefecture. In our model, we substituted the graph weighted by the

inverse distance between regions after min-max normalized for commuting data. The closest distance's edge weight is given as 1, and smaller values indicate longer distances. We term this model with inverse distance data (DD) between the 47 prefectures "GCN+Seq2seq w/ DD."

## Evaluation metrics

Two evaluation metrics were used to compare each model's predictive performance: coefficient of determination $R^2$ and mean absolute error (MAE). The $R^2$ coefficient represents how well the predicted values conform to true values; the higher, the better. The MAE is the average magnitude of differences between the predicted and true values; the lower, the better.

## Settings

We predicted influenza epidemics in the 47 prefectures of Japan with a spatiotemporal model. The model was validated as follows. The influenza patient numbers from week 1 to week 5 ("Nowcasting" and "Forecasting") were predicted using the proposed model, GCN+Seq2seq w/ PF, and the five models for comparison, i.e., VAR, LSTM, CNNRNN-Res, GCN+Seq2seq w/ AD, and GCN+Seq2seq w/ DD. We assessed the predictive performance using data from four flu seasons in Japan (31st week of 2016—30th week of 2017, 31st week of 2017—30th week of 2018, 31st week of 2018—30th week of 2019, 31st week of 2019—30th week of 2020); these were year-long periods. We set 156 weeks (three years) as the training period using past data, and then set 52 weeks (one year) as the validation period for the prediction interval estimation before each testing period. In other words, we used 7332 training samples (156 weeks × 47 prefectures), 2444 validation samples, and 2444 test samples.

We used the influenza data for 26 weeks before a specific week as inputs for all models, except the VAR model, for which we set $T'$ as five weeks. The L2-regularization of the VAR model was searched from the set of (0.01, 0.1, 1) in the validation period. Moreover, we used two hidden layers in the LSTM. The size of the hidden layer was selected as (5, 20, 50, 80, 150, 200) for the validation period. For CNNRNN-Res, the hidden dimension for the RNN was (5, 10, 20, 40), and the number of residual links was selected as (4, 8, 16), as described in [6]. For GCN+Seq2seq w/ PF, w/ DD, and w/ AD, we set the number of diffusion steps $K$ as 3. We subsequently selected the learning rate and hidden layer sizes of the GRU, $M$ and $S$, as (0.001, 0.01, 0.1, 1.0) and (32, 64, 128, 256) for the validation period, respectively. During training, all model parameters were updated using gradient descent with the Adam update rule, with a dropout value of 0.5. The dropout was applied to hidden layers to avoid overfitting and estimate model uncertainty.

# Results and discussions

## Experimental results

The results are presented in Table 2. Our GCN+Seq2seq w/ PF model outperformed all other models in terms of *MAE* and $R^2$ when predicting the number of influenza patients two to five weeks in advance.

In immediate-future predictions, such as one or two weeks in advance, the predictive performance of **VAR**, a statistical model, had no significant difference from that of the machine-learning model. However, when predicting more than three weeks in advance, the performance of the statistical model declined sharply. **LSTM**, a neural network, achieved high $R^2$ and *MAE* values considering temporal dependency. **CNNRNN-Res**, which combines a CNN and RNN using prefecture information, also achieved high performance similar to that of LSTM. However, the prediction performances of the two comparative models based on machine

**Table 2. Regional prediction model performances (averaged across all 47 prefectures in Japan).**

| Season | Model | 1-week | | 2-week | | 3-week | | 4-week | | 5-week | |
|---|---|---|---|---|---|---|---|---|---|---|---|
| | | *MAE* | *$R^2$* | *MAE* | *$R^2$* | *MAE* | *$R^2$* | *MAE* | *$R^2$* | *MAE* | *$R^2$* |
| | VAR | 181.18 | 0.936 | 248.07 | 0.816 | 314.34 | 0.690 | 456.05 | 0.511 | 607.31 | 0.294 |
| 2016/31st | LSTM | 149.76 | **0.939** | 259.34 | 0.820 | 375.67 | 0.693 | 513.66 | 0.537 | 713.20 | 0.240 |
| – | CNNRNN-Res | 163.24 | 0.918 | 332.83 | 0.750 | 375.44 | 0.616 | 396.16 | 0.542 | 458.51 | 0.460 |
| 2017/30th | GCN+S2s w/ AD | 142.08 | 0.931 | 214.89 | 0.828 | 266.42 | 0.616 | 320.02 | 0.530 | 384.38 | 0.540 |
| | GCN+S2s w/ DD | **133.78** | 0.931 | 216.11 | 0.745 | 279.27 | 0.615 | 312.18 | 0.599 | 389.78 | 0.576 |
| | GCN+S2s w/ PF | 148.76 | 0.936 | **211.30** | **0.864** | **265.85** | **0.760** | **305.77** | **0.667** | **313.29** | **0.635** |
| | VAR | 237.94 | 0.902 | 420.40 | 0.781 | 699.27 | 0.362 | 663.41 | 0.131 | 987.73 | -0.396 |
| 2017/31st | LSTM | 216.64 | 0.866 | 366.58 | 0.678 | 451.27 | 0.580 | 517.39 | 0.541 | 595.47 | 0.415 |
| – | CNNRNN-Res | 210.29 | 0.891 | 343.49 | 0.733 | 440.64 | 0.621 | 498.01 | 0.532 | 610.81 | 0.423 |
| 2018/30th | GCN+S2s w/ AD | **197.20** | **0.918** | 341.72 | 0.791 | 402.10 | 0.704 | 448.38 | 0.628 | 553.52 | 0.619 |
| | GCN+S2s w/ DD | 201.53 | 0.915 | **322.87** | 0.779 | 399.76 | 0.697 | 479.11 | 0.648 | 480.35 | 0.619 |
| | GCN+S2s w/ PF | 215.31 | 0.918 | 338.03 | **0.795** | **399.42** | **0.723** | **442.31** | **0.666** | **459.12** | **0.648** |
| | VAR | 239.21 | 0.916 | 341.35 | 0.834 | 579.81 | 0.433 | 822.02 | -0.112 | 1034.90 | -0.695 |
| 2018/31st | LSTM | 167.08 | 0.912 | 263.39 | 0.815 | 310.94 | 0.620 | 368.73 | 0.673 | 417.39 | 0.562 |
| – | CNNRNN-Res | 168.01 | 0.917 | 375.27 | 0.652 | 422.63 | 0.528 | 512.72 | 0.437 | 615.29 | 0.400 |
| 2019/30th | GCN+S2s w/ AD | 130.31 | 0.967 | 237.31 | 0.907 | 266.94 | 0.882 | 290.93 | 0.852 | 362.65 | 0.737 |
| | GCN+S2s w/ DD | 146.09 | 0.961 | 256.22 | 0.895 | 306.38 | 0.859 | 335.03 | 0.830 | 398.36 | 0.745 |
| | GCN+S2s w/ PF | **117.40** | **0.974** | **196.45** | **0.918** | **228.85** | **0.884** | **230.85** | **0.887** | **224.62** | **0.890** |
| | VAR | 126.56 | 0.942 | 326.09 | 0.420 | 544.50 | -0.803 | 686.64 | -1.901 | 856.59 | -3.459 |
| 2019/31st | LSTM | 124.62 | 0.842 | 263.39 | 0.581 | 286.15 | 0.409 | 369.28 | 0.188 | 433.53 | -0.316 |
| – | CNNRNN-Res | 100.91 | 0.922 | 283.58 | 0.571 | 333.94 | 0.357 | 399.31 | 0.151 | 501.02 | -0.402 |
| 2020/30th | GCN+S2s w/ AD | 83.95 | 0.955 | 193.15 | 0.667 | 274.65 | 0.395 | 345.11 | 0.095 | 408.42 | -0.255 |
| | GCN+S2s w/ DD | 96.82 | **0.959** | 209.93 | 0.704 | 313.72 | 0.472 | 395.97 | 0.164 | 447.38 | -0.074 |
| | GCN+S2s w/ PF | **76.26** | 0.954 | **164.05** | **0.707** | **227.57** | **0.473** | **288.72** | **0.230** | **343.89** | **0.006** |

learning at more than four weeks ahead were insufficient. **GCN+Seq2seq w/ PF**, based on a GCN using commuting data, had a slightly better prediction performance for the number of influenza patients at one and two weeks as compared with the other models; it especially achieved much better performance for predictions more than four weeks in advance. These results indicate that our GCN model based on commuting data was the best model among various epidemic prediction models for regions and countries.

## RQ1: Effectiveness of commuting data

To answer RQ1 (Does commuting data improve the accuracy of the influenza prediction?), we compared GCN+Seq2seq w/ PF with the GCN+Seq2seq w/ AD and GCN+Seq2seq w/ DD models, which used adjacency and distance data instead of commuting data, respectively. GCN+Seq2seq w/ PF outperformed both variants. Specifically, as shown by the comparison between GCN+Seq2seq w/ PF and other baselines, such as the LSTM and CNNRNN-Res models, the effect of the commuting data on advanced predictions (such as the four or five week prediction) was higher than that on immediate future predictions (such as the one week prediction). The results demonstrate the advantages of considering PF between prefectures to improve predictions of the numbers of patients that might be affected by infectious diseases. Such flow and movement of people leads to the spread of influenza from person to person.

Fig 3 shows examples of trained filters by GCN+Seq2seq w/ AD, w/ DD, and w/ PF centered at the Nara prefecture. The weights represent the importance of using inputs from other

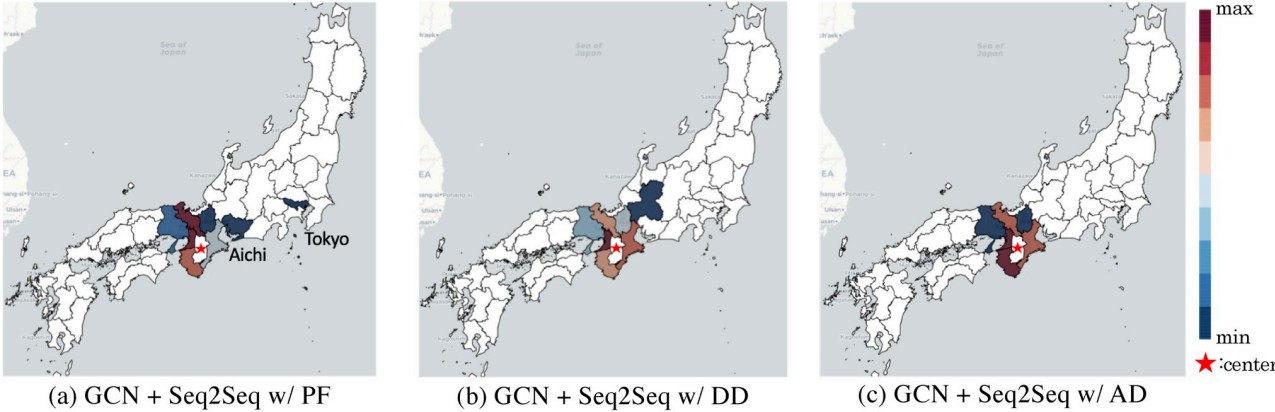

**Fig 3. Visualization of the weights of learned localized filters of Eq (2) for (a) GCN+Seq2seq w/ PF, (b) GCN+Seq2seq w/ DD, and (c) GCN +Seq2seq w/ AD against the prediction target node (Nara prefecture, as shown by a star).** The colors represent the weights, i.e., strength of influence of each prefecture on the prediction of the target prefecture. The red prefectures are given assigned larger weights, i.e., they contribute significantly for to predicting the epidemics of in the target prefecture, while blue prefectures are given assigned smaller weights. Note that most prefectures are represented in white for visibility, as their weights are less than 5% of the maximum.

prefectures. Moreover, the weights by GCN+Seq2seq w/ PF reflect commuting data, as opposed to w/ AD and w/ DD. For example, the visualization by GCN+Seq2seq w/ PF indicates significant weights for Osaka (second-largest metropolitan prefecture in Japan) and relatively significant weights for Tokyo (capital of Japan) and Aichi (third-largest metropolitan prefecture in Japan), although these prefectures are far from Nara.

## RQ2: Effectiveness of spatiotemporal model

To answer RQ2 (when and in which areas does our model produce good results?), we divided it into two questions: "for which areas does our model produce good results" and "when does our model produce good results?"

**For which areas does our model produce good results?.** For almost all prefectures, our model outperformed LSTM in terms of *MAE*. The model also provided a better performance over a wider space. We demonstrated the improvement of our model's predictive performance compared with LSTM in terms of *MAE* in the best five and least five improved prefectures, as shown in Table 3. Their locations are presented in Fig 4. These results demonstrate that

**Table 3. Improvement percentage of our predictive performance compared with LSTM in terms of *MAE* in the five most and least improved prefectures.** Lower values indicate greater improvement because a lower *MAE* indicates better performance.

| Rank | 2016/31st–2017/30th | | 2017/31st–2018/30th | | 2018/31st–2019/30th | | 2019/31st–2020/30th | |
|---|---|---|---|---|---|---|---|---|
| | Prefecture | Improve ment (%) | Prefecture | Improve ment (%) | Prefecture | Improve ment (%) | Prefecture | Improve ment (%) |
| 1 | Tokushima | -79.5 | Aomori | -46.1 | Oita | -50.7 | Kochi | -61.1 |
| 2 | Kagawa | -75.0 | Nigata | -45.5 | Gunma | -48.2 | Kagoshima | -60.5 |
| 3 | Hiroshima | -74.0 | Fukui | -39.9 | Okayama | -47.8 | Wakayama | -60.4 |
| 4 | Okayama | -69.8 | Ishikawa | -39.6 | Ehime | -47.6 | Miyazaki | -58.7 |
| 5 | Yamaguchi | -66.9 | Toyama | -38.7 | Kagawa | -47.2 | Saga | -55.7 |
| 43 | Gifu | -34.0 | Okinawa | -14.9 | Shizuoka | -17.3 | Akita | 9.1 |
| 44 | Shiga | -32.3 | Kyoto | -12.5 | Tokyo | -14.5 | Hukushima | 10.6 |
| 45 | Fukushima | -27.8 | Kochi | -11.6 | Okinawa | -13.1 | Nagano | 14.4 |
| 46 | Yamagata | -24.2 | Okayama | -11.5 | Yamaguchi | -5.0 | Aomori | 15.9 |
| 47 | Okinawa | -17.5 | Shiga | -9.1 | Miyazaki | 1.6 | Hokkaido | 20.5 |

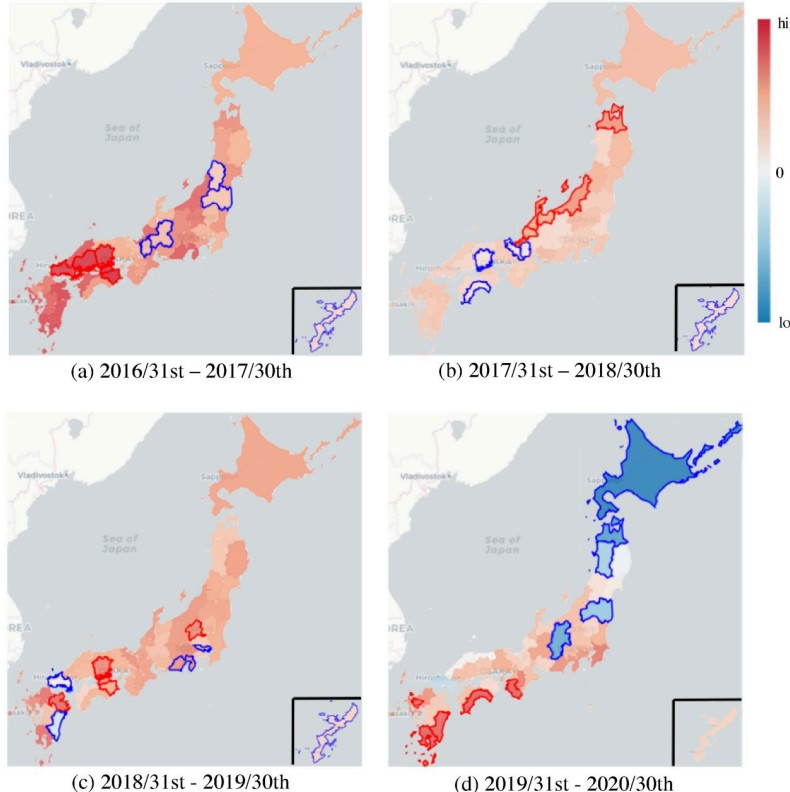

(a) 2016/31st − 2017/30th

(b) 2017/31st − 2018/30th

(c) 2018/31st - 2019/30th

(d) 2019/31st - 2020/30th

**Fig 4. Prefecture maps that illustrate the improvements of prediction accuracy measured by *MAE* in each prefecture, where the improvement ratios of GCN+Seq2seq w/ PF against LSTM are represented by colors.** Red denotes improved prefectures and blue denotes degraded prefectures. Prefectures enclosed in red and blue frames denote the five best and worst prefectures in each year, respectively. The small square at the corner of each map shows Okinawa prefecture.

GCN+Seq2seq w/ PF had a strong positive effect, such as maximizing the reduction in *MAE* by up to approximately 80%, for prediction of influenza patient numbers in any prefecture. The flow of people between different prefectures was the main factor that improved the accuracy of infection predictions.

The next important question we want to address is "what factors lead to different results of our model, compared with LSTM, in different prefectures?" Fig 4 reveals a strong relationship between locations of top-ranked prefectures (enclosed in red frames). These include four prefectures in 2016–2017, four in 2017–2018, and three in 2018–2019, which are contiguous. Hence, the GCN ensures a synergistic effect between contiguous regions. In contrast, Fig 4 reveals that the locations of prefectures with the lowest ranks (blue frames) are unrelated, except for Okinawa. Okinawa has the lowest rank of improvement (*MAE*) compared with LSTM for almost every year. We assumed that this is due to the location of Okinawa, which is the southernmost prefecture and is surrounded by sea (rightmost island in Fig 4), implying that few commuters travel there from other prefectures. Therefore, the GCN does not affect the improvement of the predictive performance for Okinawa as much as other prefectures.

**When does our model produce good results?.** Fig 5 shows the time series for Okayama with a relative *MAE* improvement compared with LSTM in four years. According to these results, GCN+Seq2seq w/ PF can identify the beginning of epidemics in specific regions. This is because it uses the GCN to learn the effects of influenza epidemics from other prefectures.

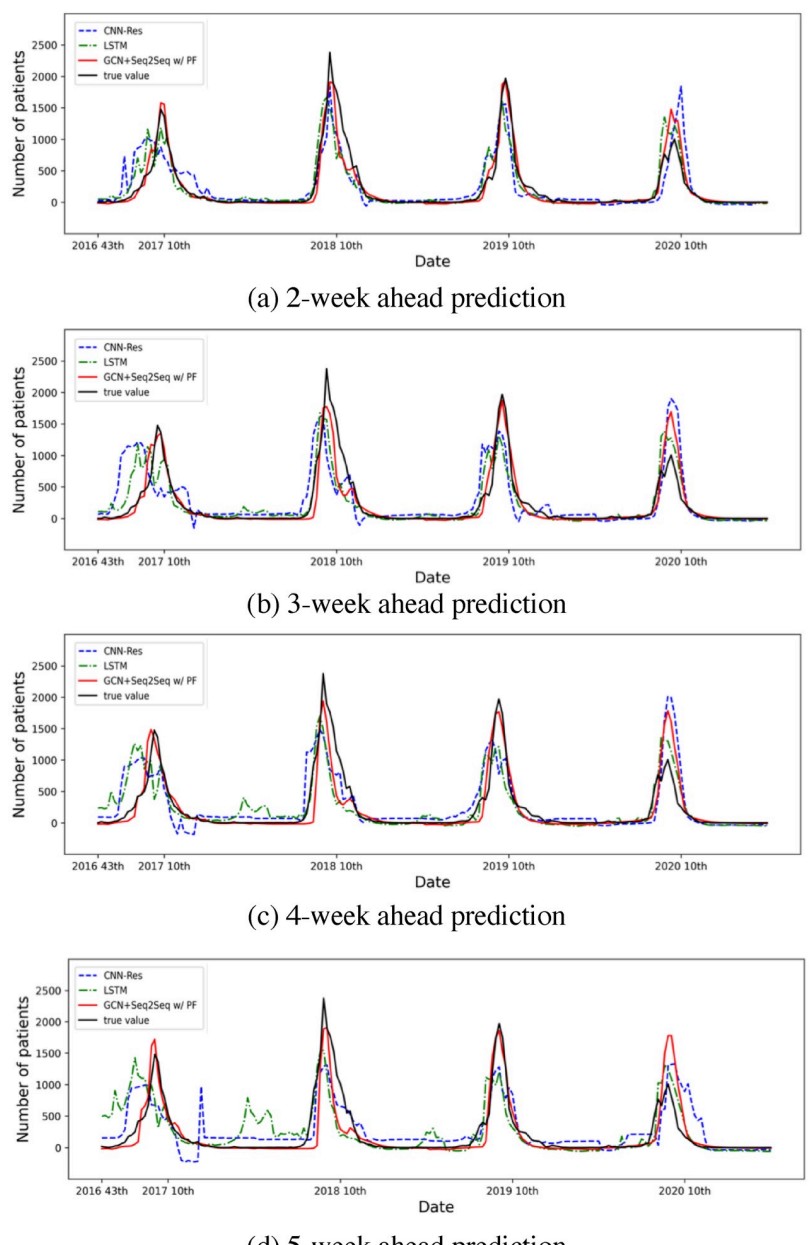

(a) 2-week ahead prediction

(b) 3-week ahead prediction

(c) 4-week ahead prediction

(d) 5-week ahead prediction

**Fig 5. Time series for Okayama prefecture: (a) two weeks in advance, (b) three weeks in advance, (c) four weeks in advance, and (d) five weeks in advance prediction time series in Okayama.** The blue and green dotted lines indicate the prediction values of compared models. The red line indicates the prediction values of the proposed GCN+Seq2seq w/ PF model. The black Line indicates the actual influenza patients.

All model predictions were lower than the true values at the peak of trends in 2018. In contrast, the results for 2020 seem to show the inverse; all model predictions were higher than the true values at the peak of trends. We assume that this tendency is due to the characteristics of machine-learning methods, which are designed to learn the data of most recent years. Evidently, in the seasons when epidemics grew much larger than in the previous years (as in 2018), these prediction models tended to underestimate the peak value. Furthermore, for

seasons when the epidemics remained on a smaller scale than in the previous years, the models overestimated the peak value (as in 2020).

## RQ3: Effectiveness of the proposed prediction interval estimation method

We evaluated the quality of our interval estimation method for epidemic prediction and compared it with Zhu's method to answer RQ3 (How effective is our uncertainty estimation method in real-world epidemic prediction?) We measured the average bandwidth, which indicates the number of patients included between the upper and lower limits of the prediction interval. We set the empirical coverage of the 95% prediction interval of each method as the validation of the prediction interval quality. This method aimed to provide good interval estimation, with a narrow average bandwidth and high empirical coverage. To search for a suitable window width $W$, we attempted to use various values (1, 3, 5, 7) in the experiment.

The results are presented in Table 4. The proposed method reduced the average bandwidth mark by 25%–32% compared to the conventional method; the empirical coverage was approximately 85%–91%, compared with that of Zhu's method, which was approximately 89%–91%. These results demonstrate the effectiveness of our proposed method. Regarding the search for a suitable window width $W$, the average bandwidth and empirical coverage tended to increase as the window width increased. The value of the window width should be determined based on the problem characteristics. This is because there is a trade-off between the average bandwidth and empirical coverage. In this scenario, a window width ($W$) of 5 caused a 29%–34% reduction in the average bandwidth and approximately 1% reduction in the empirical coverage compared with those in Zhu's method. Therefore, we assumed that a window width of 5 was sufficient.

We present a time series with a prediction interval using the proposed method in Fig 6(b); the settings are the same as shown in Fig 6(a) using the conventional method in our model. The prediction interval's width in Fig 6(b) decreases in a non-epidemic period when the true values do not escape from the interval; this increases the epidemic period. This study demonstrated applications of the proposed method to infection epidemics. Furthermore, this method can be useful for other periodic time series (such as traffic and sales volume). However, a shortcoming of this method is the requirement for periodicity in terms of validation. For example, for the application of the proposed method to influenza prediction, we require at least one year of validation data because the data have a periodicity of one year. This leads to the possibility of more validation data being required than in Zhu's method.

**Table 4. Average bandwidth and empirical coverage of the 95% prediction interval found using the proposed method and Zhu's method.**

|  |  | Zhu's | Proposed (numbers correspond to $W$) | | | |
| --- | --- | --- | --- | --- | --- | --- |
|  |  |  | 1 | 3 | 5 | 7 |
| Average band width | 1-week | 966.64 | 619.76 | 652.87 | 675.33 | 696.09 |
|  | 2-week | 1472.79 | 918.57 | 974.97 | 1016.89 | 1054.74 |
|  | 3-week | 1798.09 | 1069.76 | 1136.20 | 1190.87 | 1240.26 |
|  | 4-week | 2051.89 | 1224.23 | 1302.95 | 1371.27 | 1434.28 |
|  | 5-week | 2151.01 | 1351.16 | 1443.91 | 1529.80 | 1610.46 |
| Empirical coverage (%) | 1-week | 91.14 | 88.97 | 90.37 | 90.79 | 91.40 |
|  | 2-week | 90.07 | 85.96 | 88.87 | 89.11 | 89.49 |
|  | 3-week | 90.00 | 86.34 | 87.11 | 88.68 | 88.85 |
|  | 4-week | 89.89 | 86.45 | 86.70 | 88.45 | 89.29 |
|  | 5-week | 89.87 | 85.30 | 86.26 | 88.23 | 88.38 |

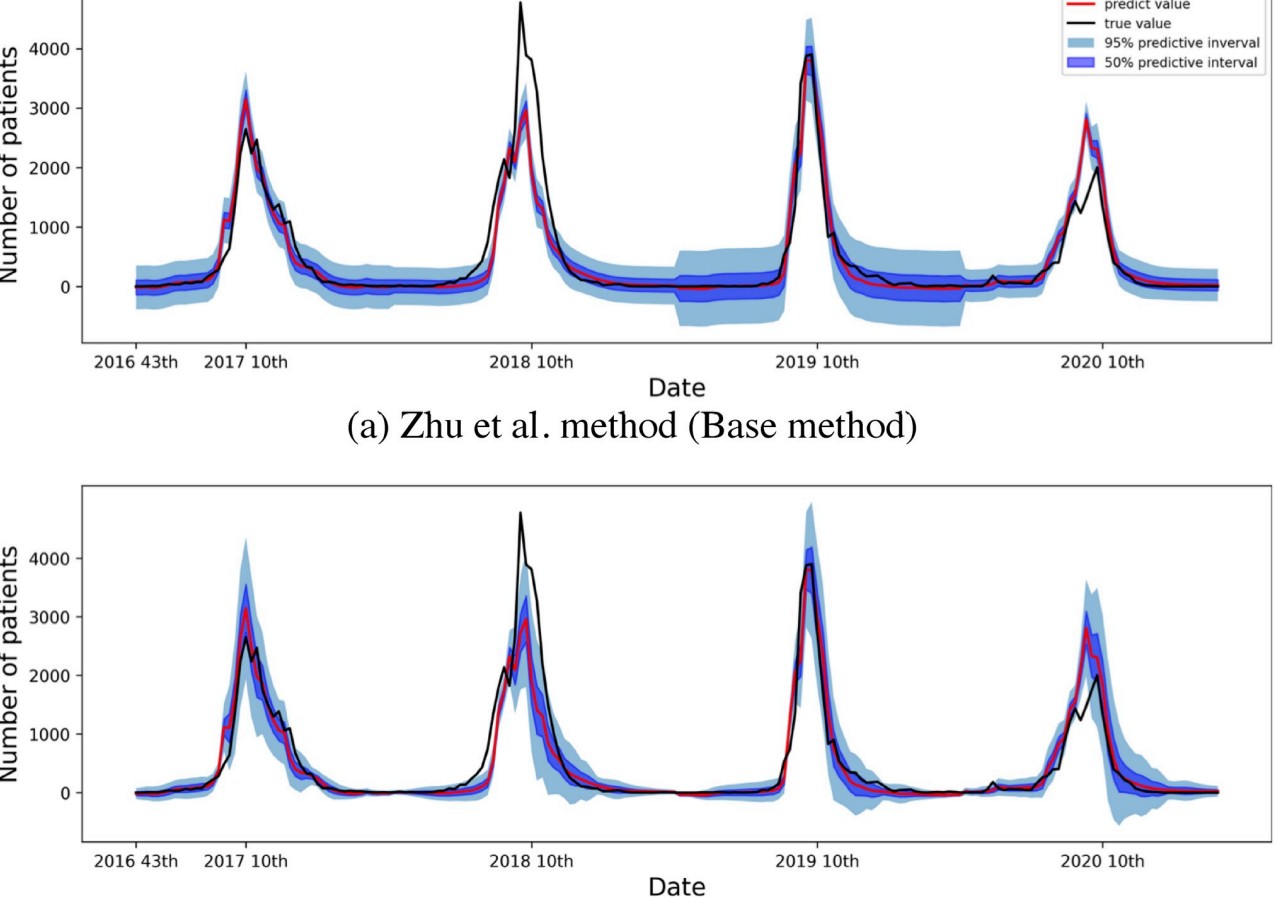

**Fig 6. Time series with prediction interval of influenza patients (black line).** Predictive values of the two weeks in advance prediction by our proposed model (red line) in the Okayama prefecture. Prediction intervals by (a) Zhu's method and (b) proposed method. Light blue and dark blue sections show the 95% and 50% prediction intervals, respectively.

## Conclusion

This study proposed a model for regional influenza prediction with uncertainty estimation by incorporating commuting data between regions. We conclude by emphasizing the following points: (1) We validated the use of PF as spatial information in a GCN for epidemic prediction. Our GCN-based model outperformed other baseline models. To the best of our knowledge, this is the first study to apply a GCN model to an epidemic prediction problem. (2) We proposed an uncertainty estimation method for periodic time series data, which reduced the prediction interval bandwidth.

The proposed model with uncertainty estimation will contribute to the infection control measures of public health organizations. Nevertheless, more research could be conducted; specifically, future work can examine the use of user-generated content in neural networks to elucidate the dynamics of other geographically evolving epidemics.

## Supporting information

**S1 Fig. Boxplots of the distribution of the prediction scores in each prefecture.** This figure shows the boxplots of the distribution of the prediction scores ($MAE$ and $R^2$) in each

prefecture for the compared models. Each colored box indicates a different model; from left to right: VAR (cyan), LSTM (green), CNNRNN-Res (blue), GCN+S2s w/ AD (pink), GCN+S2s w/ DD (brown), and GCN+S2s w/ PF (red). The black center line in each box indicates the median value; the top and bottom of each box indicate the upper and lower quartiles, respectively; the whiskers indicate the maximum and minimum values; and the other points indicate outliers. For visualization, only MAE scores from 0 to 2000 and $R^2$ scores from -1.0 to 1.0 are shown.
(TIF)

## Author Contributions

**Conceptualization:** Taichi Murayama.

**Writing – original draft:** Taichi Murayama.

**Writing – review & editing:** Nobuyuki Shimizu, Sumio Fujita, Shoko Wakamiya, Eiji Aramaki.

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
