## [Decision Letter · Decision Letter 0]

23 Sep 2020

PONE-D-20-20268

Predicting Regional Influenza Epidemics with Uncertainty Estimation using Commuting Data in Japan

PLOS ONE

Dear Dr. Murayama,

Thank you for submitting your manuscript to PLOS ONE. After careful consideration, we feel that it has merit but does not fully meet PLOS ONE’s publication criteria as it currently stands. Therefore, we invite you to submit a revised version of the manuscript that addresses the points raised during the review process.

We look forward to receiving your revised manuscript.

Kind regards,

Tzai-Hung Wen, Ph.D.

Academic Editor

PLOS ONE

Journal Requirements:

2. Please provide more information on the data used in your study. Specifically please report the following in your methods section:

     - Influenza data: date range of influenza cases chosen for this study, the date the data was accessed, the identification numbers of the entries or link to where the data can be found

     - Commuting data: the date the data was accessed, the identification numbers of the entries or link to where the data can be found

In addition, in your methods section and ethics statement, please clarify whether all data were fully anonymized before you accessed them.

"This study was supported in part by Yahoo Japan Corporation.

The funders had no role in study design, data collection and analysis, decision to publish, or preparation of the manuscript"

We note that one or more of the authors have an affiliation to the commercial funders of this research study : Yahoo Japan Corporation.

3.1. Please provide an amended Funding Statement declaring this commercial affiliation, as well as a statement regarding the Role of Funders in your study. If the funding organization did not play a role in the study design, data collection and analysis, decision to publish, or preparation of the manuscript and only provided financial support in the form of authors' salaries and/or research materials, please review your statements relating to the author contributions, and ensure you have specifically and accurately indicated the role(s) that these authors had in your study. You can update author roles in the Author Contributions section of the online submission form.

3.2. Please also provide an updated Competing Interests Statement declaring this commercial affiliation along with any other relevant declarations relating to employment, consultancy, patents, products in development, or marketed products, etc.  

4. We note that Figure 1, 2, 3, 4 in your submission contain map images which may be copyrighted. All PLOS content is published under the Creative Commons Attribution License (CC BY 4.0), which means that the manuscript, images, and Supporting Information files will be freely available online, and any third party is permitted to access, download, copy, distribute, and use these materials in any way, even commercially, with proper attribution. For these reasons, we cannot publish previously copyrighted maps or satellite images created using proprietary data, such as Google software (Google Maps, Street View, and Earth). For more information, see our copyright guidelines: http://journals.plos.org/plosone/s/licenses-and-copyright.

4.1.    You may seek permission from the original copyright holder of Figure 1, 2, 3, 4 to publish the content specifically under the CC BY 4.0 license. 

4.2.    If you are unable to obtain permission from the original copyright holder to publish these figures under the CC BY 4.0 license or if the copyright holder’s requirements are incompatible with the CC BY 4.0 license, please either i) remove the figure or ii) supply a replacement figure that complies with the CC BY 4.0 license. Please check copyright information on all replacement figures and update the figure caption with source information. If applicable, please specify in the figure caption text when a figure is similar but not identical to the original image and is therefore for illustrative purposes only.

Reviewers' comments:

Reviewer's Responses to Questions

**Comments to the Author**

1. Is the manuscript technically sound, and do the data support the conclusions?

Reviewer #1: Partly

Reviewer #2: Yes

2. Has the statistical analysis been performed appropriately and rigorously? 

Reviewer #1: No

Reviewer #2: Yes

3. Have the authors made all data underlying the findings in their manuscript fully available?

Reviewer #1: Yes

Reviewer #2: Yes

4. Is the manuscript presented in an intelligible fashion and written in standard English?

Reviewer #1: Yes

Reviewer #2: No

5. Review Comments to the Author

Reviewer #1: 1. The reference number order in the article is messy, please renumber in numerical order.

2. In lines 265 to 269, the author made the following explanation for Influenza data: "We use data for the weekly number of patients with influenza symptoms...of clinical information". However, Influenza's disease transmission and virus patterns are very complicated, and there may be asymptomatic infections, which are important parameters that cannot be captured in the data. The following points should be further explained:

(a) Which subtype of Influenza is the simulation study for?

(b) How to set the parameters that meet this Influenza subtype, especially the E (Exposed) infected persons in the SEIR model in the compartmental models?

(c) Generally, people infected with Influenza are very likely to be infected again after recovery. How to correctly evaluate the number of people infected after recovery?

(d) Since this study is based on Influenza, it does not seem to be based on a certain subtype. However, when all Influenza subtypes have been mixed and discussed, how to verify the experimental results with actual data? (Especially the data of E infected persons cannot be obtained)

3. The author uses the commuting network (Fig.1) to simulate the spread of disease in this study. However, generally speaking, the commuting network may use different means of transportation such as highways, trains, high-speed rails, planes, and ships. Sometimes the disease may spread on a large scale because an infected person takes a plane or a ship, and the plane and ship play a long-distance transmission role. However, the communicating network of Fig.1 seems to be too simple, and there is no network layer for airplanes and ships, but the simulation result (Fig.5) is very close to the true value. Can the author further explain the information acquisition and use of the commuting network in this study?

4. The results of this study are shown in Fig.4~6 until October 2019. Can the author provide a simulation to the first half of this year?

5. The number of references in 2019 and 2020 is very small (only 1 out of 50), please add the reference for the past two years.

Reviewer #2: The study proposed a graph convolutional network (GCN) based prediction model to predict influenza epidemic, which temporal trend has a 'periodicity' pattern. The model incorporate commuting data (from 2015 census) and spatial adjacency relationship as the interaction between (47) areas in the model, and used 3 flu seasons (from 2016 to 2019) as the study periods, to test and compare their model with other methods. With the three research questions analyses, they concluded that their proposed model outperform the other previous models. While the idea, method and analyses are interesting, the current status of the manuscript is yet to reach publishable quality. Therefore, I would recommend major revision. My concerns were listed as follow.

Major concerns:

1. One key contribution of the study should be the consideration of 'periodicity in a time series' (page 2, line 51) in the model, which was neglected in the previous Zhu et al. [18] Encoder-Decoder model. But, the authors did not explain what is it, and why it is important. Since the study did not use time-dependent dynamic commuting data, I would 'guess' the periodicity is in the weekly disease data. The authors should not let readers to guess, thus they should explain and clarify the 'periodicity' term where they first mention it, and emphasize the consequence of neglecting it; and which would help emphasizing the contribution of this study.

2. According to the dataset description, the commuting data is in 'the daily average number of commuters from one area to another area'. Is this dataset differentiate weekdays/weekends, or from Monday to Sunday? The time unit for the model is by weekly basis, how did the daily data converted to weekly before the 'min-max normalization'?

3. Also about the description of the commuting data (page 9, line 276-282), the authors describe the number of commuters as 'inflow of commuting data', e.g. the 270,000 and 135,000, as the number of commuters from one area to another. Based on the terminology from graph theory and social network analysis, the term 'inflow' could indicate the total number of people/commuters go to a target area, e.g. the total number of people go into Tokyo from any area; and the counterpart 'out-flow' could mean the total people leaving from the area. The usage of term 'inflow' is misleading.

4. Following the #3 point, the input data for the model should be a weighted directed matrix (as suggested in figure 1). Commuting data is expected to be the number of people commute from the home area to work area. The people eventually will go back to their home in daily basis, i.e. a reversed direction flow relationships, or transpose matrix of flow matrix. Why the reverse direction of commuting flow is not considered and processed in the model? And, why direction of flow matters in the machine-learning based model?

5. Both figures 5 and 6 suggested that all models' predictions were lower than the true values at the peak of trends, especially the second peak (near 2018 10th). Why they all failed to capture the peak values? Why LSTM's peaks were almost all earlier than the true value, whereas CNN-Res were always later?

6. In page 3 line 91, the authors claimed that 'Our study is the first to predict the influenza volume in detail on a large area...'. But in fact, the model considered only 47 areas, which is not a large number and is a low resolution for the whole country. Practically speaking, the 47 areas (assumably prefectures) might be enough for national level management, but they are too large for local disease control or health management, therefore not so useful for 'regional public health organizations' (page 15 line 459). Is this model applicable to smaller areas (higher resolution, e.g. municipal)? If so, what should be prepared and which part should be modified; if not, why?

7. Following previous point, is it possible to extend/apply the model to be used in early warning system?

8. From the view of spatial epidemiology, the disease spread from one place to another, through droplets or direct/indirect physical interactions (etc.) and through the flow of the infected people. The infectious process is described as SIR model, which has (at least) three conditions: susceptible, infected, and recovered. The infected person go through the SIR process, and thus a time-lag is expected in the process, i.e. from susceptible to infected, and from infected to recovered. How does this machine-learning based model(s) handle the complicated SIR (or SEIR, SLIR, SIS, etc.) process and the time-lag effect?

Suggestions and minor concerns:

1. Table 2 presented the average MAE and R-squared of 47 areas. While the average values shows that their model (GCN+S2s w/ PF) are in overall outperform other models, the average values may be misleading by outliers. Thus, showing the distribution of the 47 values were needed, e.g. with std or boxplots. I believe these results could be presented using a set of boxplots (3 MAE and 3 R-squared), with vertical axis showing the MAE or R-squared, horizontal-axis showing the 1-to-5-weeks, and six boxes (different colors) for each week showing the values for 47 areas for the six models. Line plot with error bars can also be used to show the average and plus-minus standard deviation if boxplot is not clear.

2. Following previous point, it should be possible to calculate the MAE and R-squared in aggregated (national) level, instead of average of 47, and the national level results shall also be useful for discussion.

3. The comparative model (GCN+S2s w/ AD) considered only the adjacent relations between areas (polygon shapes of the 47 areas). In transportation and spatial analysis, the strength of interaction between cities (e.g. flows) can be estimated mainly using gravity model or radiation model. In simple words, the interaction strengths were higher between closer cities, and lower between farther cities, i.e. distance decay effect. What if adding another comparative model that calculate the inversed distance as the weight matrix?

4. Page 12 line 378, what is 'examples of learned'. What do the colors means in Figure 3. Tokyo and Aichi were in the 'min' values, which should means that the interactions from Nara to Tokyo/Aichi are low, thus not important and could be ignored?

5. Figure 4, consider adding legends on maps. And since the authors presented a map for a year, it would be better to show the 'improvement' percentage in all 47 areas with color ramp, and maybe used colored thick borders to highlight the highest and lowest five areas.

6. Figures 3 and 4, what is the purpose of the small squares at the corner of each map (not the Hokkaido area)?

7. Table 1, consider align the second column (Definitions or Descriptions) to left.

8. Figures 5 and 6 consider changed x-label to Weeks from Date.

9. Finally, while the content is quite rich, the English writing in the manuscript is not publishable; some of the sentences needs to read twice or more to understand/guess the authors meaning, e.g. the above point 4 (examples of learned?), page 13 line 416 (beginning of epidemics?). It would be difficult for readers to understand the idea/method/uniqueness/contribution of the study, and possibly lead to misunderstanding. Please revise the writing.

6. PLOS authors have the option to publish the peer review history of their article (what does this mean?). If published, this will include your full peer review and any attached files.

Reviewer #1: No

Reviewer #2: **Yes: **WEI CHIEN BENNY CHIN

---

## [Author Response · Author response to Decision Letter 0]

24 Nov 2020

We appreciate the time and effort you and each reviewer has dedicated to providing insightful feedback on ways to strengthen our paper. We have incorporated changes that reflect the detailed suggestions you have graciously provided. We also hope that the edits and responses we have provided satisfactorily address all the issues and concerns you and the reviewers have noted.

We describes our replies to all comments in our rebuttal letter in the attached files. We kindly ask for your confirmation.

---

## [Decision Letter · Decision Letter 1]

28 Jan 2021

PONE-D-20-20268R1

Predicting Regional Influenza Epidemics with Uncertainty Estimation using Commuting Data in Japan

PLOS ONE

Dear Dr. Murayama,

Thank you for submitting your manuscript to PLOS ONE. After careful consideration, we feel that it has merit but does not fully meet PLOS ONE’s publication criteria as it currently stands. Therefore, we invite you to submit a revised version of the manuscript that addresses the points raised during the review process.

We look forward to receiving your revised manuscript.

Kind regards,

Tzai-Hung Wen, Ph.D.

Academic Editor

PLOS ONE

Reviewers' comments:

Reviewer's Responses to Questions

**Comments to the Author**

1. If the authors have adequately addressed your comments raised in a previous round of review and you feel that this manuscript is now acceptable for publication, you may indicate that here to bypass the “Comments to the Author” section, enter your conflict of interest statement in the “Confidential to Editor” section, and submit your "Accept" recommendation.

Reviewer #2: All comments have been addressed

Reviewer #3: All comments have been addressed

2. Is the manuscript technically sound, and do the data support the conclusions?

Reviewer #2: Yes

Reviewer #3: Yes

3. Has the statistical analysis been performed appropriately and rigorously? 

Reviewer #2: Yes

Reviewer #3: Yes

4. Have the authors made all data underlying the findings in their manuscript fully available?

Reviewer #2: Yes

Reviewer #3: Yes

5. Is the manuscript presented in an intelligible fashion and written in standard English?

Reviewer #2: Yes

Reviewer #3: Yes

6. Review Comments to the Author

Reviewer #2: (No Response)

Reviewer #3: In this revised manuscript, the authors have answered the reviewer accordingly. Apparently, this version is significantly improved. As an additional reviewer, I would like to provide some extra suggestion.

1. The graph element is very crucial in this study, and thus relevant information should be given as clear as possible. For example, why is the diffusion graph (Pg5) needed while the graph information has already given (Pg 9, “Commuting Data”). Is the diffusion process is inherence process of GCN or else?

Furthermore, it seems that there are only single (cross-sectional) commuting data, since the articles states “…provides only the number of commuters, regardless of the year” (pg 9, “Commuting Data” section). Is that mean such information used throughout the GCN model, or as initial information and subsequently evolve through the diffusion process? Such information would be helpful for those readers not familiar in GCN.

2. Recently, some study (see reference) also applied geographically weighted regression (GWR) into epidemic prediction. The reason that I raise this suggestion is that GWR also considers the spatial flow relation between regions which is similar in this study. This study may indicates GWR-based method may be improved using commuting data. Adding such information may be helpful for those researchers who using “statistical and time series” approach.

Reference:

Liu, F., Wang, J., Liu, J., Li, Y., Liu, D., Tong, J., Li, Z., Yu, D., Fan, Y., Bi, X., Zhang, X., & Mo, S. (2020). Predicting and analyzing the COVID-19 epidemic in China: Based on SEIRD, LSTM and GWR models. PloS one, 15(8), e0238280. https://doi.org/10.1371/journal.pone.0238280

https://www.ncbi.nlm.nih.gov/pmc/articles/PMC7451659/

7. PLOS authors have the option to publish the peer review history of their article (what does this mean?). If published, this will include your full peer review and any attached files.

Reviewer #2: No

Reviewer #3: No

---

## [Author Response · Author response to Decision Letter 1]

5 Mar 2021

Thank you for inviting us to submit a revised draft of our manuscript (PONE-D-20-20268).

We appreciate the time and effort you and each reviewer have dedicated to providing insightful feedback to help strengthen our manuscript. Thus, it is with great pleasure that we resubmit our manuscript for further consideration. We have incorporated changes that reflect the detailed suggestions you have graciously provided.

We have included our response to the reviewer in a separate file (Plos_one_reply.pdf).

Would you check the file.

Thank you.

---

## [Decision Letter · Decision Letter 2]

7 Apr 2021

Predicting Regional Influenza Epidemics with Uncertainty Estimation using Commuting Data in Japan

PONE-D-20-20268R2

Dear Dr. Murayama,

We’re pleased to inform you that your manuscript has been judged scientifically suitable for publication and will be formally accepted for publication once it meets all outstanding technical requirements.

Kind regards,

Tzai-Hung Wen, Ph.D.

Academic Editor

PLOS ONE

Additional Editor Comments (optional):

Reviewers' comments:

Reviewer's Responses to Questions

**Comments to the Author**

1. If the authors have adequately addressed your comments raised in a previous round of review and you feel that this manuscript is now acceptable for publication, you may indicate that here to bypass the “Comments to the Author” section, enter your conflict of interest statement in the “Confidential to Editor” section, and submit your "Accept" recommendation.

Reviewer #3: All comments have been addressed

2. Is the manuscript technically sound, and do the data support the conclusions?

Reviewer #3: Yes

3. Has the statistical analysis been performed appropriately and rigorously? 

Reviewer #3: Yes

4. Have the authors made all data underlying the findings in their manuscript fully available?

Reviewer #3: Yes

5. Is the manuscript presented in an intelligible fashion and written in standard English?

Reviewer #3: Yes

6. Review Comments to the Author

Reviewer #3: The authors have made their points clear and convincing. I have no further question. Note that this paper has illustrated an advanced method for disease modelling and thus related details should be stated correctly. Please ensure no typo mistake, notation mistake during the publication process.

7. PLOS authors have the option to publish the peer review history of their article (what does this mean?). If published, this will include your full peer review and any attached files.

Reviewer #3: No

---

## [Editor Report · Acceptance letter]

13 Apr 2021

PONE-D-20-20268R2 

Predicting Regional Influenza Epidemics with Uncertainty Estimation using Commuting Data in Japan 

Dear Dr. Murayama:

I'm pleased to inform you that your manuscript has been deemed suitable for publication in PLOS ONE. Congratulations! Your manuscript is now with our production department. 

Kind regards, 

on behalf of

Dr. Tzai-Hung Wen 

Academic Editor

PLOS ONE